# Probiotics in Treatment of Viral Respiratory Infections and Neuroinflammatory Disorders

**DOI:** 10.3390/molecules25214891

**Published:** 2020-10-22

**Authors:** Roghayeh Shahbazi, Hamed Yasavoli-Sharahi, Nawal Alsadi, Nafissa Ismail, Chantal Matar

**Affiliations:** 1Cellular and Molecular Medicine Department, Faculty of Medicine, University of Ottawa, Ottawa, ON K1H 8M5, Canada; rshah017@uottawa.ca (R.S.); hyasa068@uottawa.ca (H.Y.-S.); nalsa068@uottawa.ca (N.A.); 2School of Psychology, Faculty of Social Sciences, University of Ottawa, Ottawa, ON K1N 6N5, Canada; Nafissa.Ismail@uottawa.ca; 3School of Nutrition, Faculty of Health Sciences, University of Ottawa, Ottawa, ON K1H 8M5, Canada

**Keywords:** probiotics, immunomodulation, gut microbiota, gut-lung axis, viral respiratory infections, COVID-19, influenza virus infection, gut-brain axis, neuroinflammation, multiple sclerosis

## Abstract

Inflammation is a biological response to the activation of the immune system by various infectious or non-infectious agents, which may lead to tissue damage and various diseases. Gut commensal bacteria maintain a symbiotic relationship with the host and display a critical function in the homeostasis of the host immune system. Disturbance to the gut microbiota leads to immune dysfunction both locally and at distant sites, which causes inflammatory conditions not only in the intestine but also in the other organs such as lungs and brain, and may induce a disease state. Probiotics are well known to reinforce immunity and counteract inflammation by restoring symbiosis within the gut microbiota. As a result, probiotics protect against various diseases, including respiratory infections and neuroinflammatory disorders. A growing body of research supports the beneficial role of probiotics in lung and mental health through modulating the gut-lung and gut-brain axes. In the current paper, we discuss the potential role of probiotics in the treatment of viral respiratory infections, including the COVID-19 disease, as major public health crisis in 2020, and influenza virus infection, as well as treatment of neurological disorders like multiple sclerosis and other mental illnesses.

## 1. Introduction

The human intestine is inhabited by diverse and dynamic microbial communities known as gut microbiota [1]. Firmicutes, Proteobacteria, Actinobacteria, and Bacteroidetes are the predominant phyla in the gastrointestinal tract with the highest densities in the colon [2]. The Gut microbiota has important biological functions, including regulation of host metabolism and immune system homeostasis, maintenance and improvement of the function of the intestinal epithelial barrier, and prevention of pathogens invasion [1]. The establishment of the gut microbiota begins at birth and before the third year of life, infant microbiota resembles the gut microbiota of adults [3]. Although gut microbiota composition is relatively stable in adults, lifestyle factors can alter this stability and result in dysbiosis [1,3]. Dysbiosis is defined as any alteration in the composition of the commensal microbial population living in the human gut relative to the population found in healthy individuals [4]. Various factors including diet, certain medications, and exposure to toxins and pathogens can contribute to a dysbiosis state [5]. Gut dysbiosis is associated with inflammatory disorders, including inflammatory bowel diseases, allergy, asthma, obesity, metabolic syndrome, type 1 and type 2 diabetes, and central nervous system (CNS)-related disorders [5,6].

The largest compartment of the immune system belongs to the intestine which is constantly in contact with a variety of antigens and immunomodulatory components derived from the diet and the commensal microbiota [7]. Approximately 70% of the immune system is found at the level of the intestinal epithelial barrier. Therefore, the intestinal epithelial barrier and its related immune system play a key role in maintaining the homeostasis of the immune system and mounting a protective response against pathogens and inflammation [8]. Microbial communities in the intestine contribute to the development, maintenance, and function of the immune system. The gut microbiota influences immunity by the maturation of gut-associated lymphoid tissue and innate lymphoid cells, enhancing antimicrobial peptides expression such as regenerating islet-derived protein γ and β by intestinal epithelial cells, increasing IgA-producing B cells expression, promoting antibodies, and cytokines production, and inducing T cells differentiation [9,10,11,12]. Taken together, gut microbial imbalance negatively influences gut immunity, systemic immunity, and distant organs immunity such as lungs and brain and leads to inflammation [13,14]. 

Inflammation is a biological response to the immune system activation by the various stimuli, including infectious agents such as bacterial or viral infections, or non-infectious agents, including tissue injury, cell death, toxic compounds, and degeneration [15,16]. These stimuli induce major inflammatory signaling pathways, which trigger acute and chronic inflammatory responses in different organs and cause tissue damage and disease [16]. A large body of research has shown that intake of natural products such as probiotics may display a protective role against inflammatory disorders [17,18]. 

Probiotics are known as natural products, which confer health benefits on the host [19]. Probiotics have been suggested to prevent or treat gut microbial dysbiosis through maintenance of the integrity of the intestinal barrier and immunomodulation activity [20,21]. Probiotics maintain intestinal barrier integrity through inhibiting pathogens growth and colonization by competition for nutrients and attachment sites on epithelial cells. Probiotics also play a role in reinforcing the mucosal barrier defense by inducing antimicrobial peptides and inducing mucus secretion [20,22,23]. In addition, probiotics demonstrate immunomodulation activity by increasing the frequency of immunoglobulin-A (IgA) positive cells in Peyer’s patches in the lamina propria. IgA inhibits bacterial adherence to epithelial cells and neutralizes toxins [22]. Further, probiotics modulate gut adaptive immune responses. These microorganisms downregulate the production of T-helper17 (Th17) inflammatory cells and their signature pro-inflammatory cytokine, IL-17F, as well as decrease IL-23 production which is essential for the expansion, stabilization, and function of Th17 [22,24,25]. Probiotics also reduce the production of tumor necrosis factor-alpha (TNF-α) which is a downstream Th-17-related cytokine [22,26]. Some probiotics can upregulate T regulatory cells (Tregs) production; therefore, probiotics play a crucial role in inhibiting inflammatory responses and regulating immune cell homeostasis [23,27,28].

Given the growing evidence suggesting the role of probiotics in the treatment of inflammatory disorders through affecting gut microbiota [29,30], we review the potential role of probiotics in treatment of viral respiratory diseases by affecting the gut-lung microbiota axis, focusing on COVID-19 and influenza infections. We also discuss the impact of probiotic intake on neurological disorders by affecting the gut-brain microbiota axis focusing on multiple sclerosis (MS) and mental illnesses. 

## 2. Effect of Probiotics on Gut-Lung Axis and Viral Respiratory Infections

### 2.1. Gut-Lung Axis and Viral Respiratory Infections

In addition to inducing inflammatory conditions at the gut level, gut dysbiosis affects the respiratory tract and leads to inflammatory diseases, such as asthma, allergy, and chronic obstructive pulmonary disease (COPD) [31]. The intestine and lungs are parts of a mucosal immune system known as the gut–lung axis [32]. Immune cells travel from gut to lung through the common mucosal immune system [13]. Gut immunity imbalance due to gut microbiota perturbation may affect immune responses at the lung level [13,33]. Furthermore, gut microbial metabolites such as short-chain fatty acids (SCFAs) act as crucial local and systemic signaling molecules in maintaining immune homeostasis [34]. Gut bacteria-derived metabolites and fragments translocate from the intestinal lumen to the lungs through the mesenteric lymphatic system and systemic circulation and subsequently may induce lung immune responses [13,35,36,37,38,39]. Interestingly, the interplay between gut and lung is bilateral because inflammation in the lung can also cause changes in gut microbiota [13]. 

Acute respiratory infections, especially viral respiratory infections are one of the main causes of morbidity and mortality among children and adults worldwide and lead to the development of other diseases [40]. Commensals-derived signals and metabolites protect the host against respiratory infections, as disturbance or absence of microbiota in animal models may be accompanied by deteriorated immune responses and exacerbated outcomes following bacterial or viral pulmonary infections due to impaired innate and adaptive immunity [31,41,42]. Gut bacteria regulate the activation of the interferons (IFNs) signaling, which is crucial for response against the majority of viruses, and inflammasome activity, an innate signaling pathway that is involved in defense against a subset of viruses [42]. In the steady-state, gut microbial-derived signals regulate low-level signal transducer and activator of transcription-1 (STAT1) activation [42], which is critical for IFNs signaling, and is involved in antiviral defense genes induction prior to infection and immune-mediated resistance to viral infection [42,43]. Figure 1 illustrates the role of gut microbiota in the prevention of viral respiratory infections.

Some evidence demonstrated that probiotics bacteria may reduce the incidence of bacterial and viral infections of airways [31,44,45]. In animal models, antibiotic-induced changes of gut microbial composition have been shown to increase predisposition to respiratory diseases and airways viral infections, while some species of *Lactobacillus* and *Bifidobacterium* reduced the incidence and improved the outcomes of airways viral infection [31]. Probiotics exert immunomodulatory activity through pattern recognition receptors such as toll-like receptors (TLR) [13]. TLRs receptors, which connect innate immunity to adaptive immunity, recognize pathogen-associated molecular patterns (PAMPs), which results in the initiation of downstream signaling cascades like nuclear factor-κB (NF-κB) [46,47]. Following the recognition of probiotics related PAMPs by TLRs, NF-κB induces expression of the antiviral gene [47]. Antigen-presenting cells (APCs), such as macrophages and dendritic cells (DCs) in the intestine, prominently express TLRs [46]. After interaction with probiotics, activated APCs induce activation of natural killer (NK) cells, which results in interferon-gamma (IFN-γ) expression and antiviral defense activation [47]. In this review, we discuss the current evidence regarding the effect of probiotics on gut microbiota and viral respiratory infections focusing on the novel coronavirus and influenza virus infections. 

### 2.2. Coronavirus Disease-2019 

The novel zoonotic pathogen Severe Acute Respiratory Syndrome Coronavirus 2 (SARS-CoV-2) is the cause of Coronavirus Disease 2019 (COVID-19). COVID-19 has been declared a global pandemic, with characteristic symptoms including extreme respiratory deterrence, pneumonia, and shortness of breath [48]. This pandemic was first identified in December 2019 in Wuhan, Hubei Province, China, and widely spread across China and beyond, resulting in a worldwide sudden and substantial increase in hospitalizations of COVID-19 patients [49,50]. Current studies report estimation of 3% for the global case-fatality rate of COVID-19, ranging from 0.4% in China to 31.4% in the northwest of Italy [51].

SARS-Cov-2 is a positive sense RNA virus with spike-like projections on its enveloped surface [50]. This virus enters the cells via the angiotensin-converting enzyme-2 (ACE2) receptors. ACE-2 receptors are also expressed in the intestinal epithelial cells [50,52]. It has been suggested that novel coronavirus infects intestinal epithelial cells via ACE2 receptors and transmembrane protease serine and induces proinflammatory chemokines and cytokines production, which leads to acute intestinal inflammation [52,53]. Furthermore, some reports indicate gastrointestinal symptoms such as diarrhea, as well as detection of SARS-Cov2 RNA in fecal samples of some patients affected by COVID-19 [50,54]. On the other hand, the gut microbial imbalance is associated with acute respiratory distress syndrome (ARDS) [50,55], and COVID-19 may progress to the acute ARDS [56]. Therefore, existing evidence may interestingly demonstrate the link between novel coronavirus and gut microbiota and involvement of the gut-lung axis in COVID-19 pathogenesis. Scientists have reported gut microbiota disruption in COVID-19 disease [57], although observations regarding the direct effect of coronavirus infection on gut microbiota are scarce [58]. Some data from China has indicated a change in the gut microbiota of some patients infected by SARS-CoV-2 toward a decrease in *Lactobacillus* and *Bifidobacterium* abundance [58]. 

Antivirals are one of the commonly administrated medications in COVID-19 patients, which may lead to gut microbiota dysbiosis; however, it is not clear whether the alteration in gut microorganism is the consequence of COVID-19 infection or is induced by treatment [59]. In a pilot study in Hong Kong, microbiomes of 15 hospitalized patients with COVID-19 were analyzed using the shotgun metagenomics technique to assay the link between microbiome alteration, disease severity, and fecal viral RNA [60]. A persistent change in the fecal microbiome was reported toward a decrease of commensals and an increase of opportunistic pathogens. An inverse correlation was observed between the abundance of *Faecalibacterium prausnitzii*, which is an anti-inflammatory bacterium, and disease severity. Moreover, there was an inverse correlation between some species from the Bacteroidetes phylum and SARS-CoV-2 fecal load. Changes in gut microbial composition persisted even after virus clearance [60]. Microbiome functional analysis revealed elevated functional capacity for nucleotide and amino acid biosynthesis and carbohydrate metabolism [61]. Therefore, modulation of the gut microbiota might be a beneficial strategy to reduce the gastrointestinal outcomes and severity of the disease [60]. Probiotics can be administrated both to prevent gut microbiota imbalance due to exposure to predisposing conditions such as some medications or diseases, and as a therapeutic strategy to rebuild the gut microbial ecosystem [62]. 

On the other hand, probiotics influence lung microbiota as well [63] and exert anti-inflammatory activity in the lungs [64]. For instance, oral administration of probiotic *Lactobacillus helveticus* positively modulated the immune system [65,66,67,68] and had an immunoprotective impact on mucosal immunity by increasing the number of cells secreting IgA in the gut and bronchial-associated lymphoid tissue [69]. Importantly, in the case of COVID-19, it is now accepted that the local innate immune response, especially secretory IgA, is a main defensive mechanism in the early phases of infection [70]. Evidence from influenza virus and beta coronavirus infections characterizes immunopathological features of these diseases, including impaired acute inflammatory responses involving macrophages, neutrophils, DCs, TLRs, cytokines, and CD4+ and CD8+ T-cells [71]. As mentioned before, probiotics regulate innate and adaptive immune systems through modulating TLRs pathways and regulating the activity of DCs and NK cells, modulating T-helper 1 (Th1) and T-helper 2 (Th2) mediated responses, and maintaining CD4+ and CD8+ T-cells balance [22,72]. The positive effect of probiotics on the prevention of upper respiratory tract infection development, reduction in disease duration and severity, and improvement of outcomes has been demonstrated in both adults and children [33,73]. 

Patients with COVID-19 have higher serum concentrations of cytokines, including TNF-α, IFN-γ, IL-2, IL-4, IL-6, and IL-10 [74]. A high level of proinflammatory cytokines is associated with severe complications such as ARDS or septic shock in these patients [75]. Therefore, effective treatment to reduce inflammation-related lung damage and severe complications of COVID-19 is critical. The results from a recent meta-analysis of a randomized clinical trial (RCT) showed that supplementation with probiotics can mitigate inflammation in adults by decreasing serum level of pro-inflammatory cytokines, including hs-CRP, TNF-a, IL-6, IL-12, and IL-4 [76].

Respiratory failure and hypoxemia occur in some patients with COVID-19, and those patients may need invasive ventilation [77]. An RCT in critically ill patients has reported that the administration of symbiotic (a mixture of probiotics and prebiotics) decreased days of stay in the intensive care unit and days under mechanical ventilation [33]. Ventilator-associated pneumonia (VAP) is a serious complication of mechanical ventilation [77]; several RCTs have revealed the beneficial effect of probiotics therapy in the prevention of VAP in critically ill patients [58].

Taken together, modulation of gut microbiota by probiotics is suggested as a possible strategy to improve clinical manifestation of COVID-19 [50]. There is no direct data about the effectiveness of probiotics on the prevention or treatment of coronavirus infection, although the antiviral activity of some strain of probiotics against other coronaviruses or other viral respiratory infections has been detected [53]. Therefore, the direct impact of probiotics administration on COVID-19 outcomes remains to be determined by conducting animal studies and/or clinical trials.

### 2.3. Influenza Virus Infection

Influenza is considered as one of the most common types of acute respiratory infections and a major cause of mortality [78,79]. Acute respiratory infections stimulate antiviral immunity that is related to alteration of microbial composition at both lungs and intestine levels [78]. Research has demonstrated that gut microbiota plays an influential role in shaping defense against influenza virus infection and improving outcomes of the disease [80]. Influenza infection can be recognized by the innate immune system through the TLRs, including TLR7 and the retinoic acid-inducible gene I (RIG-I). Following infection of airway epithelial cells by the influenza virus and activation of TLRs pathway, APCs such as DCs prime effector T cells responses. Subsequently, influenza virus-specific CD4 and CD8 T cells promote viral clearance in the lungs [79]. In influenza virus infection, the production of cytokines such as proinflammatory cytokines (IL-1α, IL-1β, IL-6, TNF-α), IL-4, IL-10, and IFNs are critical for mounting the host-immune defense to ameliorate symptoms [81]. Experimental studies indicate that commensal microbiota regulates the production of virus specific CD4 and CD8 T cells and antibody responses upon influenza virus infection, whereas depletion of gut bacteria by antibiotic treatment increases the susceptibility to influenza infection [82]. 

Animal experiments and human studies provide insight into the beneficial effect of probiotics against pulmonary viral infections like influenza partially by modulating gut microbiota and gut immunity [80]. In the murine model, administration of different strains of *Lactobacillus*, *Bifidobacterium*, and *Lactococcus* could ameliorate the symptoms of the influenza infection [83].

Takeda et al. (2011) observed that oral administration of 10 strains of lactic acid bacteria isolated from traditional Mongolian dairy products to the influenza-infected mice alleviated infection symptoms by its immunomodulatory activity through intestinal immunity [81]. Feeding mice with probiotics elevated gene expressions of the IL-12 receptor and IFN-γ in Peyer’s patches and augmented NK cells activity and Th1 related cytokines production via intestinal immunity [81]. Further, oral administration of heat-killed *Lactobacillus gasseri TMC0356* could increase mRNA expression of IL-12, IL-15, and IL-21 in Peyer’s patches, decrease virus titers in lung, and ameliorate clinical symptoms in mice [84].

In a study by Yitbarek et al. (2018) the effect of probiotics administration on the prevention of influenza infection was assayed in chickens with antibiotic-induced gut dysbiosis [85]. Animals were fed with a mixture of five *Lactobacillus* species. Higher virus shedding and lower expression of type I IFNs and IL-22 (a member of the IL-10 family of cytokines which is associated with barrier function maintenance and innate antimicrobials stimulation at mucosal surfaces [86]) were reported in antibiotic-treated chickens compared to controls, whereas probiotics treatment could decrease virus shedding and elevate IL-22 expression to a normal level [85]. Furthermore, feeding influenza-infected mice with a human isolate of *Bifidobacterium longum MM-2* decreased inflammatory responses in the lower respiratory tract and mortality through activation of NK cells in the lungs and spleen and upregulating of cytokines expression in the lungs [87]. Similar immunomodulatory activity and protective effect against influenza virus have been reported in murine models using probiotics *Bifidobacterium longum BB536* [88], *Lactobacillus rhamnosus* (*L. rhamnosus*) [89], *Lactobacillus paracasei* [90], *Lactobacillus brevis KB290* [91], *Lactobacillus acidophilus* L-92 [92], *Lactobacillus plantarum* [93], *Lactobacillus fermentum* [94], and *Lactobacillus kunkeei* [95].

The effect of probiotics supplementation in the prevention of influenza infection has been demonstrated in human studies as well; however, we could not find any clinical trial analyzing the gut microbiota in relation to the effect of probiotics on influenza virus infection. A study revealed that consumption of a probiotic drink containing *Lactobacillus brevis KB290* led to a reduction in the incidence of influenza in schoolchildren, although microbiome analysis was not done in this study [96]. 

Aging leads to the change in gut microbial composition, as a decline in some gut microbial communities such as *Bifidobacterium*, *Faecalibacterium*, *Akkermansia*, and *Clostridium cluster XIVa* has been observed in the elderly population [71]. Age-related changes in the gut microbial population negatively affect the immune system in the elderly population [71]. Several clinical trials have proposed that modulation of the gut microbiota using probiotics may improve immune responses to infections and influenza vaccination in elderly subjects [97,98].

In contrast, all probiotic strains might not improve protection against respiratory virus infections. Some studies in elderly subjects have shown no significant effect of probiotics on respiratory infections [99]. In a RCT conducted in 737 healthy elderly people, daily intake of *Lactobacillus casei Shirota* (*L. casei Shirota*) for 176 days did not decrease the occurrence of respiratory symptoms and did not improve the influenza-vaccination immune response in the group receiving probiotic compared to the control group [100]. Similar findings have been reported in other age groups. In a study of 523 children aged 2–6 years, daily consumption of a milk beverage containing *L. rhamnosus GG* for 28 weeks did not decrease the incidence of various respiratory virus infections, including influenza virus, and respiratory symptoms [101]. In addition, intake of yogurt fermented with *Lactobacillus delbrueckii ssp. bulgaricus OLL1073R-1* did not show a significant preventive effect against influenza in women healthcare employees [102]. Therefore, besides the strain of administrated probiotic, other factors such as characteristics of participants may affect the efficacy of probiotics against respiratory infections.

Together, most existing evidence supports the anti-inflammatory and immunomodulatory activities of probiotics against influenza infection, partly through modulation of gut microbiota; however, there are limited clinical trials in this field.

## 3. Effect of Probiotics on the Gut-Brain axis and Neuroinflammation

### 3.1. Gut-Brain Axis and Neuroinflammation

There is a bilateral interaction between the gut and brain known as the gut-brain axis [103]. The communication in the gut-brain axis comprises many direct and indirect pathways. Gut microbiota influences these routes of communication and exerts a significant effect on the CNS, brain neurochemistry, and activity [104,105]. Gut microbiota communicates with CNS, mainly by the secreting of microbial products, including peptidoglycans, SCFAs, and tryptophan metabolites such as serotonin. These metabolites can influence the brain immunity either by inducing signaling pathways (e.g., neuroimmune signaling) or by entering circulation and crossing the blood-brain barrier (BBB) [104,106]. For instance, SCFAs have been shown to play an important role in the development and function of microglia [107]. Microglia are the brain’s macrophages, which abundantly express TLRs and are involved in antigen presentation, phagocytosis, and modulating inflammation [104,106]. Besides, gut microbes and their metabolites can improve BBB integrity by increasing the expression of tight junction proteins and decreasing its permeability [108]. BBB dysfunction has been observed in some neuroinflammatory disorders [109]. Studies involving germ-free or antibiotic-treated mice have reported an adverse impact on neurodevelopment and neurodegenerative disorders due to the disruption of gut microbiota signaling to the CNS involved in neuroimmune development [104,110]. Figure 2 illustrates the role of gut microbiota in the prevention of neuroinflammation.

On the other hand, CNS affects gut microbiota composition and function directly, through the secretion of endocrine mediators, namely, catecholamines, and indirectly, by affecting the gut environment through the autonomic nervous system that modulates the gut physiology, gut barrier integrity, and permeability [104,111,112]. 

Numerous studies have shown a correlation between gut dysbiosis and progression of neuroinflammatory diseases, including MS [113,114,115], mental illnesses such as depressive and anxiety-like behavior [116,117], Parkinson’s disease [118,119,120], and Alzheimer’s disease [121,122,123]. Modulation of gut microbiota by probiotics may promote immune response, mitigate inflammation, and reduce the progression of neurological disorders [124]. Next, we review the scientific evidence concerning the influence of probiotics intake on MS and mental illnesses through modulation of commensal gut bacteria. 

### 3.2. Multiple Sclerosis 

MS is an inflammatory disease of the CNS that leads to a decrease in BBB integrity, inflammatory cells infiltration of perivascular tissues, myelin sheath degradation, and axonal damage. The main immunological changes are the elevated expression of proinflammatory CD4+ T cells (Th1 or Th17 cells), monocytes, macrophages, inflammatory DCs, and B cells, reduction in CD8+ T cells and FoxP3+ Tregs expression, and Tregs dysfunction [125,126]. Both genetic and environmental factors such as vitamin D deficiency, antibiotic exposure, obesity, chemical and biological pollutants, and gut microbiota dysbiosis are involved in the etiology of MS [125]. Some differences have been observed in the gut microbiota composition of people with MS compared to healthy subjects [127]. Alteration in gut microbiota and microbial metabolites leads to an imbalance between Th17 cells and Tregs populations and induces inflammatory pathways, which in turn affect the autoimmune response in the CNS [35,128]. Further, gut commensals exert protective activity against MS by maintaining the integrity and function of the gut barrier. Following gut barrier damage, translocation of bacterial derivatives from the gut lumen to the systemic circulation increases, which in turn may induce myelin-reactive T cells in both peripheral lymphoid tissues and the brain [129]. Probiotics consumption and their beneficial metabolites can restore gut microbiota, maintain the gut barrier, regulate immune cells homeostasis, and mitigate chronic inflammation [35,128]; therefore, they may be useful preventive and therapeutic strategies against inflammatory and autoimmune diseases, including MS [130].

Several studies have investigated the potential role of probiotics in the treatment and alleviating MS symptoms in experimental models or clinical settings. Animal studies using the experimental model of MS (experimental autoimmune encephalomyelitis (EAE)) have revealed the significant role of probiotics treatment on regulating the immune system homeostasis via inhibition of Th1/Th17 expression and induction of Tregs expression [131,132,133,134]. For example, in a study treating mice with a mixture of five different probiotics (*Lactobacillus casei, Lactobacillus acidophilus, Lactobacillus reuteni, Bifidobacterium bifidum,* and *Streptococcus thermophilus*) before induction of EAE, suppressed EAE development, delayed disease onset, reduced the pro-inflammatory Th1/Th17 polarization, and increased IL10+ and Foxp3+ Tregs expression in the peripheral immune system and inflammation site [135]. Similarly, in another study treating mice with a mixture of five strains of *lactobacilli* suppressed the progression of EAE, inhibited pro-inflammatory Th1 and Th17 related cytokines expression, and induced expression of IL-10+ and Foxp3+CD4+CD25+ Tregs [136]. 

There is limited research investigating the effect of probiotics on the gut microbiome and EAE; although existing evidence supports the efficacy of probiotics in ameliorating the severity of EAE symptoms through restoring gut microbiota, and modifying the immune system and inflammatory responses. Feeding rats with probiotic *Candida kefyr* modified gut microbiota, decreased Th17 and IL-6 production and induced Tregs, and CDl03-positive regulatory DCs production in the intestine, which caused a reduction in the severity of EAE symptoms [137]. He et al. (2019) reported that *Lactobacillus reuteri* (*L. reuteri*) administration inhibited Th1/Th17 cells expression and their related cytokines IFN-γ/IL-17 in EAE mice and prevented EAE progression. They also discovered *L. reuteri* treatment significantly restored gut microbial diversity. A negative correlation was observed between *Bifidobacterium, Prevotella*, and *Lactobacillus,* and EAE severity, while the genera *Anaeroplasma*, *Rikenellaceae*, and *Clostridium* were positively related to EAE severity. Interestingly, *L. reuteri* intake modified the relative frequency of these taxa [138]. Chen et al. (2019) examined the effect of treatment of mice with either probiotic *Clostridium butyricum* (*C. butyricum*), a butyric acid-producing bacterium, (3 weeks before EAE induction) or antibiotic norfloxacin (one week before EAE induction) on gut microbiota and immune response in EAE mice [139]. Despite differences in gut microbiota composition between groups, both treatments alleviated EAE. The abundance and diversity of the gut microbiota were increased in *C. butyricum*-treated EAE mice while decreased in norfloxacin group; although both interventions reduced the firmicutes/bacteroidetes ratio. In both groups, a decrease in Th17 response and an increase in Tregs response were observed [139].

The impact of probiotics on restoring gut microbiota and regulating immune system responses in preclinical model of MS was well documented. In the clinical setting, Tankou, et al. (2018) have reported the effect of supplementation with a probiotic formulation containing *Lactobacillus*, *Bifidobacterium*, and *Streptococcus*, twice daily for 2 months, on the gut microbiome and peripheral immune function in relapsing-remitting MS patients and healthy controls [140,141]. Probiotics administration changed the relative abundance and overall microbial community structure in MS patients and controls along with a reduction in relative frequencies of Th1 and Th17 cells observed in both controls and MS patients [140]. The authors also reported enrichment of *Lactobacillus*, *Bifidobacterium*, and *Streptococcus* genera in the gut of patients and healthy controls [140,141]. These changes are well known to be associated negatively with pro-inflammatory immune markers and positively with anti-inflammatory immune markers [140,141]. 

Two studies evaluated the effect of supplementation with a probiotic capsule on clinical and inflammatory responses in patients with MS. These studies showed a decrease in inflammatory markers [142,143]; Overall, existing data support the beneficial effect of probiotics on inhibiting MS progression, reducing disease severity, and modifying immune-inflammatory responses, but limited studies have focused on gut microbiota. It seems clinical studies are required to further decipher the interaction between probiotics, gut microbiota, and immune system response in MS.

### 3.3. Mental Illnesses

The gut microbiome plays a significant role in neural development, cognition, and behavior [144]. Dysbiosis and inflammation of the gut are related to several mental illnesses such as anxiety and depression [145,146]. Gut inflammation increases gut permeability and leads to bacterial translocation and release of cytokines and neurotransmitters. Therefore, these molecules can enter the systemic circulation and subsequently pass the BBB and influence brain function, resulting in anxiety and depression [116,145]. Inflammasome activation and expression of proinflammatory cytokines, such as IL-1β, IL-6, and IL-18, due to dysbiosis state, may be related to major depressive disorders [116].

Animal studies have elucidated the causative link between microbiome perturbation and depression-like behaviors, and probiotics seem to be a beneficial means to counteract the destructive effects of microbial imbalance on mental health [144,145,146]. Research has shown that probiotics are effective in the reduction of anxiety and depressive symptoms, similar to conventional prescription medications [145]. Murray et al. (2019) showed manipulation of gut microbiota with probiotics during puberty decreased LPS-induced depression-like behavior in female mice and anxiety-like behaviors in male mice later in life [147].

Antibiotics exposure is one of the main reasons for microbial dysbiosis that contributes to numerous diseases [20]. Besides depletion of pathogenic bacteria, antibiotics can kill subsets of commensal microbes as well. Thereby, antibiotics can affect gut microbiota directly by reducing the diversity and abundance of gut microbiota and indirectly by impairing symbiotic relationships among the various subset of the microbiota and changing the microenvironment in the gut, which negatively impacts on the growth of the microbial population [20]. Antibiotic intake may have long-term adverse effects on host health, as antibiotic exposure during infancy has been reported to change the abundance of specific gut bacterial populations and increases the risk of many non-communicable diseases later in life [148,149,150]. 

Existing evidence supports the role of probiotics in mitigating the deleterious effects of antibiotics on gut dysbiosis related changes in brain function [117,151]. For example, early life exposure to the antibiotic (low-dose penicillin) exerted long-term effects on gut microbiota, increased cytokines expression in the frontal cortex, and altered BBB integrity. Furthermore, the antibiotic-treated mice showed behavior alteration, including impaired anxiety-like and social behaviors, and exhibited aggression. Feeding mice with probiotic *L. rhamnosus JB-1* prevented some of the changes, including the reduction of sociability and social novelty in both male and female mice and anxiety-like behavior in females [117]. In another study, mice were treated with a mixture of broad-spectrum antibiotics, including ampicillin, streptomycin, and clindamycin for 2 weeks, and, thereafter, they were fed with probiotic *Lactobacillus casei DG* [152]. Microbiota disturbance led to general inflammation, change of some endocannabinoidome members in the gut, depression, reduced social recognition, and biochemical and functional alterations. Probiotic administration partly restored the gut microbiome, mitigated gut inflammation, and modified behavioral, biochemical, and functional alterations [152]. 

Exposure to chronic stress causes gut microbiota perturbation. The significant role of the microbiota in the development of the stress system and regulating stress-related alterations in behavior and brain function has been demonstrated [153,154]. The effect of lactic acid bacteria treatment on depression in C57BL/6J mice was examined by Tian et al. (2019) [155]. Mice were exposed to chronic stress for a period of 5 weeks and concurrently fed with *Bifidobacterium longum subsp. infantis E4* and *Bifidobacterium breve M2CF22M7*. Microbiome analysis showed that probiotics treatment effectively modified stress-mediated gut dysbiosis. It also increased the level of 5-hydroxytryptamine and brain-derived neurotrophic factor in the brain and decreased depressive behaviors [155]. Similarly, other studies using the same probiotics, found that treatment of chronically stressed mice with probiotics could restore gut microbiota perturbation, mitigate inflammation, and alleviate depression- and anxiety-like behaviors [156,157].

Furthermore, *Lactobacillus mucosae NK41, Bifidobacterium longum NK46, Lactobacillus reuteri NK33*, and *Bifidobacterium adolescentis NK98* isolated from healthy human feces have shown to exert antidepressant properties through normalization of gut microbial composition and immune system, suppressing inflammatory responses by inhibiting the NF-κB pathway, decreasing blood level of corticosterone, TNF-α, IL-6, and lipopolysaccharide levels in mice with immobilization stress-induced anxiety/depression [158,159]

In addition, some studies have reported the effectiveness of supplementation with probiotics in the reduction of anxiety and depressive symptoms in humans [160]. For instance, two studies have shown that supplementation of healthy subjects with a combination of several species of *Lactobacillus* and *Bifidobacterium* led to a significant reduction in anxiety and/or depression scores compared to control groups [161,162], none of these studies examined the gut microbial composition and gut-bran axis.

Moreover, the beneficial effect of supplementation with a probiotic on gut microbiota and anxiety symptoms in patients with chronic fatigue was reported [163]. Patients received daily *L. casei Shirota* supplement or a placebo for 2 months. Stool samples were collected and Beck Depression and Beck Anxiety Inventories were completed at the study baseline and end of the intervention. A significant increase in *Lactobacillus* and *Bifidobacteria* in stool samples of the probiotic group was reported. Therefore, investigators concluded that probiotic consumption led to the predominance of beneficial bacteria that are associated with a healthy gut. They also found a significant decrease in anxiety symptoms in the intervention group. These findings may support the role of gut microbiota in mental health [163].

## 4. Conclusions

Numerous studies highlight the significance of modulating the composition and function of gut microbiota by specific natural products like probiotics as a promising strategy to promote immune function and modulate inflammatory responses. Increasing evidence is more and more supporting the health-promoting properties of probiotics in disease models. Growing clinical evidence reveals that targeting the gut–lung microbiota axis may play a therapeutic role in viral infections such as influenza. Considering the strong links between the immunity and severity of viral infections, the development of strategies to strengthen the immune system may be effective in the prevention of SARS-CoV-2 infection or reduction of severe outcomes in patients with COVID-19 infection and management of the disease. Although probiotics have been proposed to confer beneficial effects against COVID-19 infection, the rationale for probiotics administration in SARS-CoV-2 infection prevention or treatment is originated from indirect observations. Overall, well documented probiotics for antiviral respiratory infection properties may be suggested as a therapeutic approach, along with other therapies to help reduce the disease burden and improve its outcomes.

On the other hand, regarding neuroinflammatory disorders such as MS, existing data, mostly driven from pre-clinical studies, support the role of probiotics in alleviating the severity of symptoms by affecting the get-brain microbiota axis. Scientific evidence also favors the therapeutic potentials of probiotics for mental illnesses through normalization of gut microbiota and immune responses and mitigating inflammation. However, to confirm the positive effects of probiotics on health and prevention or treatment of such diseases, and to determine the optimal strain and dose with clinical efficacy, well designed randomized controlled clinical trials should be conducted.

## Figures and Tables

**Figure 1 molecules-25-04891-f001:**
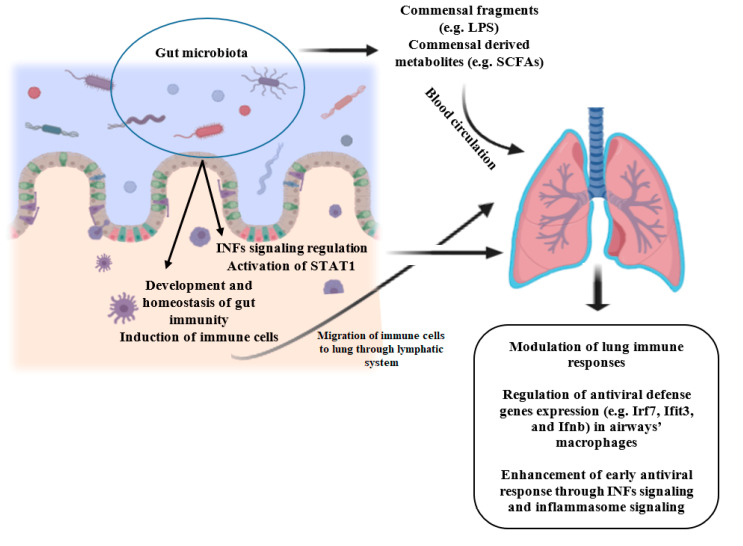
Role of gut microbiota in the prevention of viral respiratory infections. Commensals-derived fragments and metabolites travel to the lung via systemic circulation and can act as signaling molecules and induce immune responses. Furthermore, activated immune cells travel to the lung via the lymphatic system. Besides, gut bacteria regulate the activation of STAT1 and IFNs signaling, which are involved in antiviral defense genes induction prior to infection and immune-mediated resistance to viral infection. INFs: interferons; STAT1: signal transducer and activator of transcription-1; LPS: Lipopolysaccharide; SCFAs: Short-chain fatty acids. Created with BioRender.com.

**Figure 2 molecules-25-04891-f002:**
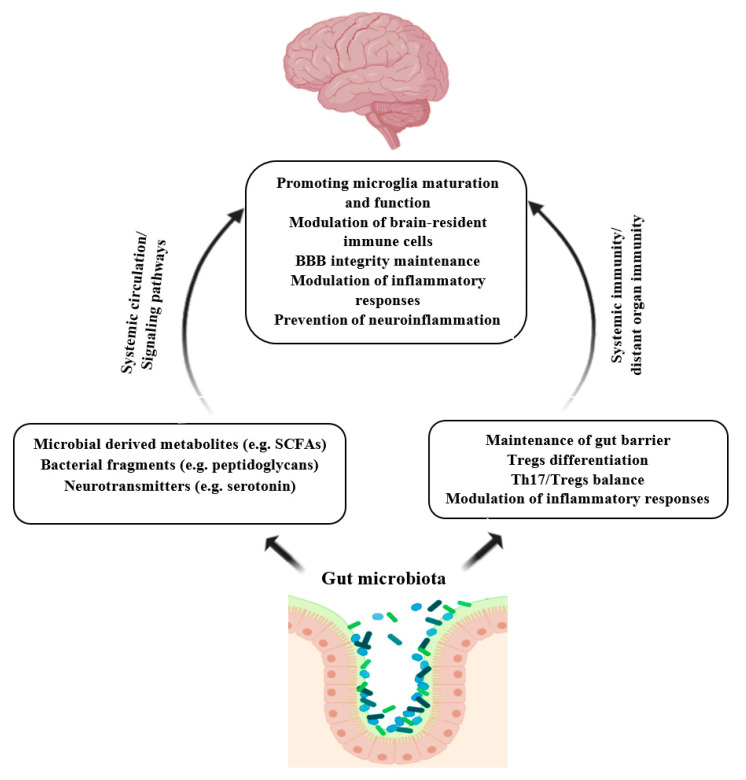
Role of gut microbiota in the prevention of neuroinflammation. Microbial products and metabolites can induce brain immunity indirectly by activating signaling pathways or directly through passing BBB. Gut microbiota regulates gut immunity hemostasis and modulates systemic immunity and brain immunity as well. Finally, gut commensal and their metabolites modulate microglia maturation and function, maintain BBB integrity, and prevent neuroinflammation. SCFAs: Short-chain fatty acids; Tregs; t regulatory cells; Th17: T-helper 17. Created with BioRender.com.

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
