# Peer review of "Probiotics in Treatment of Viral Respiratory Infections and Neuroinflammatory Disorders"

_molecules, 2020, doi:10.3390/molecules25214891_

Round 1

Reviewer 1 Report

The paper is well written and understandable. Although these features the paper does not appear to underline additional evidence nor to treat the topics from a different point of view respect to previously published reviews.
Some minor revisions, concerning the lacks of references, can be evidenced (sentences lines 37-38, 42-43)
It would be a great improvement for the paper if the authors included some considerations about the fact that not all probiotic strains appear effective in exerting antiviral activities.

Author Response

We would like to thank the reviewer for the valuable comments and their time spent on reading our manuscript.

Point 1: Some minor revisions, concerning the lacks of references, can be evidenced (sentences lines 37-38, 42-43).

Response 1: Related references were added to the sentences lines 37-38 and 42-43 (Introduction section, page 1).

Point 2: It would be a great improvement for the paper if the authors included some considerations about the fact that not all probiotic strains appear effective in exerting antiviral activities. 

Response 2: Regarding this constructive remark, related evidence was added to the "influenza virus infection section", pages 6-7, lines 267-279.

Reviewer 2 Report

The authors discuss how probiotics affect mucosal immune system in Gut-Lung Axis and gut-brain Axis and the benefits of probiotics in prevention and treatment in viral Respiratory Infections and some neuroinflammatory diseases. The review was well written. However, I have a few suggestions and concerns:

1:  In the abstract, the authors stated that “In the current paper, we discuss the therapeutic functions of probiotics on viral respiratory infections, focusing on the COVID-19 disease..”  In fact, COVID-19 is only a small part in the review and there is no direct evidence showing the effectiveness of probiotics on the prevention or treatment of coronavirus infection, therefore, the statement is misleading that readers would think the review focuses on the prevention and treatment using probiotics on COVID-19 infection.

2: The conclusion should also mention probiotics in mental illness.

Author Response

We would like to thank the reviewer for the valuable comments and their time spent on reading our manuscript.

Point 1: In the abstract, the authors stated that “In the current paper, we discuss the therapeutic functions of probiotics on viral respiratory infections, focusing on the COVID-19 disease..”  In fact, COVID-19 is only a small part in the review and there is no direct evidence showing the effectiveness of probiotics on the prevention or treatment of coronavirus infection, therefore, the statement is misleading that readers would think the review focuses on the prevention and treatment using probiotics on COVID-19 infection.

Response 1: The statement was modified as follows (page 1, lines 23-25 ):

"In the current paper, we discuss the potential role of probiotics in the treatment of viral respiratory infections, including the COVID-19 disease, as major public health crisis in 2020, and influenza virus infection"

Point 2: The conclusion should also mention probiotics in mental illness.

Response 2: the following sentence was added to the conclusion section, page 11, lines 485-487:

Scientific evidence also favors the therapeutic potentials of probiotics for mental illnesses through normalization of gut microbiota and immune responses and mitigating inflammation.

Reviewer 3 Report

The subject of the paper:  “Probiotics in Treatment of Viral Respiratory Infections and Neuroinflammation Disorders” is an emerging subject due to this Covid 19 pandemic disease. Probiotics are viable bacteria that colonize the intestine and affect intestinal microbial balance.  In this paper the authors tried to reveal that probiotic consumption may the risk or duration of respiratory infection symptoms and neuroinflammation disorders.

First studied aspect in this paper was the role o probiotics in covid disease. Acording to literature data probiotics might confer beneficial effects against COVID-19 infection, but the direct impact of probiotics administration on COVID-19 is originated from indirect observations and  outcomes remains to be determined by conducting animal studies and/or clinical trials.

Second aspect presented  is related to the role of probiotics in influenza, which is considered as one of the most common types of acute respiratory infection and a  major cause of mortality Animal experiments and human studies proved  the beneficial effect of probiotics  against pulmonary viral infections. Regarding neuroinflammatory disorders such as multiple sclerosis (MS), data driven from pre-clinical studies, support the role of probiotics in alleviating the severity of symptoms by affecting the get-brain microbiota axis.

Due the hot topic, the paper can be accepted for publications after minor correction.

References must be written in accordance with instruction for authors:

  • Journal Articles:
    1. Author 1, A.B.; Author 2, C.D. Title of the article. Abbreviated Journal Name Year, Volume, page range.

Author Response

We would like to thank the reviewer for the valuable comments and their time spent on reading our manuscript.

Point 1: References must be written in accordance with instruction for authors

Response 1: References were modified in accordance with instruction for authors